# High Precision Use of Botulinum Toxin Type A (BONT-A) in Aesthetics Based on Muscle Atrophy, Is Muscular Architecture Reprogramming a Possibility? A Systematic Review of Literature on Muscle Atrophy after BoNT-A Injections

**DOI:** 10.3390/toxins14020081

**Published:** 2022-01-21

**Authors:** Alexander D. Nassif, Ricardo F. Boggio, Sheila Espicalsky, Gladstone E. L. Faria

**Affiliations:** 1Departamento de Pesquisa, Núcleo Nassif—Ensino Médico e Pesquisa, Belo Horizonte 30411-148, Brazil; 2Departamento de Pesquisa, Instituto Boggio—Medicina Ensino e Pesquisa, Sao Paulo 04004-030, Brazil; ricardoboggio@ricardoboggio.com.br (R.F.B.); gladstonefaria@hotmail.com (G.E.L.F.); 3Departamento de Pesquisa, Clínica Sheila Espicalsky, Vila Velha 29101-104, Brazil; drasheila@sheilaespicalsky.com.br

**Keywords:** botulinum toxins, type A, botox, muscular atrophy, muscle atrophy, wrinkles, ficial lines, aesthelics, esthetics, muscular architecture reprogramming

## Abstract

Improvements in Botulinum toxin type-A (BoNT-A) aesthetic treatments have been jeopardized by the simplistic statement: “BoNT-A treats wrinkles”. BoNT-A monotherapy relating to wrinkles is, at least, questionable. The BoNT-A mechanism of action is presynaptic cholinergic nerve terminals blockage, causing paralysis and subsequent muscle atrophy. Understanding the real BoNT-A mechanism of action clarifies misconceptions that impact the way scientific productions on the subject are designed, the way aesthetics treatments are proposed, and how limited the results are when the focus is only on wrinkle softening. We designed a systematic review on BoNT-A and muscle atrophy that could enlighten new approaches for aesthetics purposes. A systematic review, targeting articles investigating BoNT-A injection and its correlation to muscle atrophy in animals or humans, filtered 30 publications released before 15 May 2020 in accordance with the Preferred Reporting Items for Systematic Reviews and Meta-Analyses (PRISMA) guidelines. Histologic analysis and histochemistry showed muscle atrophy with fibrosis, necrosis, and an increase in the number of perimysial fat cells in animal and human models; this was also confirmed by imaging studies. A significant muscle balance reduction of 18% to 60% after single or seriated BoNT-A injections were observed in 9 out of 10 animal studies. Genetic alterations related to muscle atrophy were analyzed by five studies and showed how much impact a single BoNT-A injection can cause on a molecular basis. Seriated or single BoNT-A muscle injections can cause real muscle atrophy on a short or long-term basis, in animal models and in humans. Theoretically, muscular architecture reprogramming is a possible new approach in aesthetics.

## 1. Introduction

Botulinum toxin type A (BoNT-A) has been historically used for the aesthetic treatment of facial lines. Although there are an increasing number of on-label uses to treat a variety of disorders using BoNT-A, when it comes to aesthetics, all the on-label approvals refer to facial lines [1]. Currently BoNT-A is approved by the FDA for the aesthetic treatment of forehead, glabellar, and lateral canthal lines, while in some other countries, such as Brazil, the on-label aesthetic approval is more generic and permits BoNT-A injections all over the face to treat facial lines [2,3]. The main point is that all the aesthetic on-label approvals concern facial lines only. Numerous published clinical trials objectify the improvement of facial lines after treatment with BoNT-A [4]. A multitude of articles aimed to compare the main brands of BoNT-A available on the market regarding the durability of the effect of softening wrinkles provided by these toxins [5]. Dose comparisons between BoNT-A brands generate misleading results because they are all different and are not interchangeable substances [6,7,8].

Despite differences in market brands, all currently marketed BoNT-A have one thing in common: a protein complex of 150 kDa composed of a heavy chain (HC, 100 kDa) linked via a disulfide bond to a light chain (LC, 50 kDa) [9,10,11]. After a BoNT-A injection, the simplified mechanism of action cascade can be described based on its biochemical structure [12,13,14,15,16,17] (Figure 1).

The whole cascade takes between 24 to 72 h to be completed after BoNT-A injection, and it is an irreversible process [18]. Once the SNAP-25 (synaptosomal-associated protein of 25 kDa) protein is inactivated, muscle contraction will only be reestablished after neuronal repair that depends on nerve sprouting and/or motor plate regeneration [19]. Although scientific evidence on this statement dates back to the 1970s [20], many still argue today about BoNT-A “durability” in relation to wrinkle control rather than studying the level of tissue damage caused by a BoNT-A injection and the time required for neuronal healing, as concerns aesthetics. The previous sentence is fundamental for the purpose of the new aesthetic approach of BoNT-A use in aesthetics that we intend to propose based on the real BoNT-A mechanism of action.

Many studies have demonstrated nerve terminal and nodal sprouting in the paralyzed nerves as early as two days after botulinum toxin injection [21,22]. Broadening the scope, studies on botulism have already provided a substrate to support the idea that the botulinum toxins durability for practical purposes is approximately 24 to 72 h and that the actual long-term effect of muscle paralysis depends only on nerve and muscle tissue regeneration processes. Treatment with antitoxin for patients with botulism, in order to be effective, should be started within 24 to 48 h of contamination, otherwise the already established neuronal chemical tissue injury is no longer reversible [23]. Once the disease is established by neuronal inability to release acetylcholine in the synaptic cleft of the neuromuscular junction, life support becomes essential, which is normally restricted to clinical care, with special attention to maintaining respiratory capacity, which requires mechanical ventilation for 2 to 6 months, until neuronal and muscular healing processes take place, restoring diaphragmatic and intercostal muscle function [24,25].

Studies addressing counter-terrorism measures suggest the use of antidotes against BoNT-A in the event of a mass attack using BoNT-A as a chemical weapon. Only 1 g of BoNT-A in natura is capable of decimating 1 million humans, showing that it is a powerful and lethal toxin. All of the antidotes tested, even those capable of neuronal internalization, require concern regarding the therapeutic window, which must precede a chemical neuromuscular junction denervation of 24 to 72 h [24,26].

Understanding BoNT-A’s real mechanism of action makes it possible to identify some semantic misconceptions that have been repeated historically since its first use for aesthetic purposes and that directly impact the way scientific productions on the subject are designed, the way aesthetics treatments are proposed, and how limited the results are when the focus is only on wrinkles softening. Considering the statements above and the questions raised below (Table 1), we designed a systematic review on BoNT-A and muscle atrophy that could enlighten new approaches for aesthetics purposes.

## 2. Aims

To conduct a systematic review of the literature regarding BoNT-A treatments and muscle atrophy that could support new perspectives in facial aesthetics and to propose a new reading for the aesthetic use of BoNT-A, no longer focusing on simple control of wrinkles and facial lines, but as a drug capable of selectively reprogramming long-term muscle strength and tonus through muscle atrophy. We will discuss the proposition that muscle architecture could be altered by creating areas of real atrophy—hyporesponsive or even irresponsive to acetylcholine stimuli for muscle contraction. The restoration of neuronal and muscular function would be based exclusively on the healing processes of these tissues.

## 3. Method

The present systematic review, targeting articles that investigate BoNT-A injections and its correlation to muscle atrophy in animals or humans, was conducted in a stepwise process for studies published before 15 May 2020 and in accordance with the Preferred Reporting Items for Systematic Reviews and Meta-Analyses (PRISMA) guidelines [27]. The search strategy, the flow diagram of study selection, and the data extraction are detailed below, because the review was not registered. By the time our independent research group tried to register the review at PROSPERO in 2020, we had already started article extraction. After October 2019, PROSPERO only accepted earlier registration.

STEP 1—PubMed/MEDLINE and BVS (Biblioteca Virtual em Saúde) databases were explored using the following Medical Subject Headings (MeSH) entry terms: “Botulinum Toxin Type A” OR “Botulinum A Toxin” OR “Botulinum Neurotoxin A” OR “Botox” AND combined with the MeSH entry terms “Muscle Atrophy” OR “Muscular Atrophy” (Table 2). The overlapping studies were excluded in STEP 1.

In STEP 2, the studies obtained in STEP 1 were screened by “title” and “abstract” by two independent researchers (A.D.N. and R.F.B.). Those not satisfying inclusion criteria or with exclusion criteria (Table 3) were excluded. The group of articles selected to proceed to the next step was determined through an interactive consensus process. Discrepancies were judged by a third reviewer (S.E.).

In STEP 3, the full text of all the potential articles selected in STEP 2 were obtained and carefully read to screen for those whose purposes were in accordance with the aim of the present review.

In STEP 4, the eligible studies in STEP 3 were thoroughly read, and data for each study were extracted and analyzed according to a PICO-like structured reading (Table 4).

The methodological quality of the articles included in the study was evaluated using a specific scale developed based on STROBE (Strengthening the Reporting of Observational studies in Epidemiology) principles [28]. Each item was categorized, and the maximum global score was set to 26 (Table 5).

## 4. Reults

### 4.1. Selection of the Studies

From 191 articles initially identified after removing duplicates, thirty-five were deemed relevant after reading titles and abstracts. Thirty were included in the review (5 were excluded because they did not meet the selection criteria). Sixteen were animal studies and fourteen were human studies. The PRISMA Flow Diagram of Article Selection for Review is summarized in (Figure 2).

### 4.2. Quality of the Reviewed Articles

The quality of the reviewed articles was highly variable and is summed up in Table 6 [29,30,31,32,33,34,35,36,37,38,39,40,41,42,43,44,45,46,47,48,49,50,51,52,53,54,55,56,57,58]. Most studies, 28/30, were prospective ones, with 13 well-controlled and randomized, but this subgroup was only of animal studies. The descriptive quality of the experimental protocol results, as well as their interpretations and conclusions, were adequate in most studies. The follow-up ranged from 3 months to 4 years.

### 4.3. Literature Analysis

A general overview of the population type of the 30 studies is summarized in Table 7. All Animal studies had good quality control groups. Human studies, on the other hand, lacked control groups or had poor quality control groups.

Most animal studies used mature healthy animals. Human studies, on the other hand, used very heterogeneous subjects in relation to age (varying from children to 91-year-old adults) and health status.

Overall, there were very few studies regarding the facial mimetic musculature in humans—only two: Borodic (1992) [29] and Koerte (2013) [47]. The facial masticatory musculature represented mainly by the masseter muscle were studied in three human studies: To (2001) [33], Kim (2005) [34], Lee (2007) [40]; and three animal studies: Kwon (2007) [39], Babuccu (2009) [42], Kocaelli (2016) [54].

Numerical heterogenic population samples (from 1 to 383 subjects) and qualitative heterogenic samples, more specifically in human studies (healthy and subjects with different muscle disorders), were observed.

There was also heterogenic BoNT-A dose, BoNT-A brand types used in the studies and follow-up period, summarized in Table 8.

The methodological variability among the small number of studies made it mandatory to conduct an extensive evaluation based on the identification of muscle atrophy after BoNT-A injections registered separately via different tools in animal or human studies. The general findings are summarized in Section 4.3.1. (Animal Studies) and Section 4.3.2. (Human Studies), below.

#### 4.3.1. Animal Studies

##### Muscle Balance

Muscle balance was measured in 10 out of 16 animal studies to evaluate muscle atrophy. Significant muscle balance reduction after seriated BoNT-A injections and after one single BoNT-A injection were observed in 9 out of 10 studies. The reduction varied from 18%, Fortuna (2013b) [48], to 60%, Fortuna (2011) [44], and there was a BoNT-A dose dependency/interval of injection association identified by Herzog (2007) [37], Frick (2007) [38], Tsai (2010) [43], Fortuna (2011) [44], Fortuna (2013b) [48], and Caron (2015) [51]. The higher the dose, the higher the muscle balance reduction. Long intervals between injections permitted partial muscle balance recovery. Only Fortuna (2015) [50] found no muscle balance alterations after 6 months of injection (Table 9).

##### Optical and Electron Microscopy

Hystologic (optical and electron microscopy) analysis and histochemistry showed profound muscle structure changes in animal models, such as sarcomere distortion, decrease in myofibrillar diameters, and myofibrillolysis/myonecrosis—Babuccu (2009) [42], Tsai (2010) [43], Kocaelli (2016) [54]. Significant reduction of percentage of contractile material—Frick (2007) [38], Fortuna (2011) [44], Fortuna (2013a) [45], Fortuna (2013b) [48], Fortuna (2015) [50]. Replacement of contractile fibers with fat, fatty infiltration, and increased collagen fibers forming perimysium—Herzog (2007) [37], Fortuna (2011) [44], Kocaelli (2016) [54] (Table 10).

##### Imaging

Kwon (2007) [39] showed a computed tomography (CT) scan rabbit masseter muscle volume reduction of up to 18.41% (±3.15) after 6 months of a BoNT-A injection. Magnetic resonance imaging (MRI) was used in monkeys by Han (2018) [56] and showed significant paraspinal muscles atrophy after BoNT-A injections (Table 11).

##### Molecular Biology

Direct and indirect muscle atrophy identification via molecular biology was studied and is detailed in Table 12 and Table 13.

#### 4.3.2. Human Studies

##### Optical and Electron Microscopy

Histologic (optical and electron microscopy) analysis and histochemistry showed results in humans similar to those found in animal models. Muscle atrophy (atrophic muscle fibers, myofibrillar disorganization, fibrosis, necrosis, and increase of the number of perimysial fat cells) were well-documented by Kim (2005) [34], Schroeder (2009) [41], Valentine (2016) [52], and Li (2016) [53]. The Orbicularis oculi muscle showed that the morphometric measurements of muscle fibers reduced, with an irregular diameter at 3 months after BoNT-A injections, (*p* < 0.05). Ansved (1997) [31] showed a mean diameter reduction of type IIB striated muscle fibers (Vastus lateralis) of 19.6% after 2–4 years of BoNT-A treatement (*p* < 0.05). Partial recovery of the changes described above were seen in some articles (Table 14).

##### Imaging

All the 10 human studies that evaluated images to measure muscle atrophy after BoNT-A treatments showed signs of muscle atrophy, irrespective of the technology used: ultrasound, MRI, CT scan, or cephalometry. Muscle atrophy was registered in the short term (42 days to 3 months) and in the long term (up to 2 years). No full recovery was identified (Table 15).

## 5. Discussion

The use of BoNT-A for cosmetic purposes is a fast-growing procedure, with more than six million treatments performed by plastic surgeons in the year 2018 alone [59]. Despite this significant number, we believe that improvements in BoNT-A aesthetic treatments have been jeopardized by the famous, but simplistic, statement used by the media, patients, and doctors: “BoNT-A treats wrinkles”. BoNT-A monotherapy relating to wrinkles is, at least, questionable. The BoNT-A mechanism of action is presynaptic cholinergic nerve terminals blockage by inhibition of the release of acetylcholine, causing paralysis and subsequent functional denervated muscle atrophy to some degree [60]. It is important to keep in mind that wrinkles have a multitude of causes, besides muscle contraction, and that treatments of wrinkles based only on the use of BTX-A have poor quality results in the long term [61]. Rohrich (2007) [62] brilliantly demonstrated modern topographic anatomic studies proving the relationship between wrinkles and underlying structures other than muscles, such as arteries, veins, nerves, and septa of fat compartments [62].

The use of BTX-A was first studied by Scott (1973) [63] for the treatment of strabismus by pharmacologic weakening the extraocular muscles [33]. The first described use of the toxin in aesthetic circumstances was by Clark and Berris (1989) [64], but it still carried out the essence of the BoNT-A mechanism of action based on muscle paralysis and atrophy [64]. At some point during the 1990s, Carruthers and Carruthers [65] began to use botulinum toxin type A in full-scale treatments for aesthetic purposes. Since then, the aesthetic focus regarding the use of BoNT-A moved towards removing wrinkles only [65]—a shift in the medical literature on BoTN-A for aesthetics purposes that has persisted until today. We are not underestimating the importance of Carruthers and many other authors that previously studied the use of BoNT-A in aesthetics but, as mentioned above, we intend to provide the aesthetic use of BoNT-A a new perspective. The real mechanism of actions of BoNT-A for aesthetic purposes have been forgotten, to a level where recent publications still focus on the fact that muscle paralysis and muscle atrophy is a complication of the “wrinkle treatment” capacity of BoNT-A instead of its expected effect [66,67,68].

This systematic review can shed new light on aesthetic BoNT-A treatments basing itself on old, but scientifically correct, concepts of striated muscle contraction physiology, muscle hypertrophy, and muscle atrophy—basic concepts of muscle physiology from reference physiology medical books such as the Guyton and Hall Textbook of Medical Physiology [69].

The results of this systematic review showed evidence that seriated or single BoNT-A muscle injections can cause real atrophy on a short or long-term basis, in animal models and in humans, in skeletal striated muscles of the limbs, facial masticatory muscles, and facial mimetic muscles. Due to only limited good quality data being available, we included animal model studies and human studies, but we know that data extrapolation from animal model studies to humans are, at least, naïve. The sensitivity of animals to BoNT-A has been known for many years to be less than that perceived in humans [70]. There are even differences in sensitivity between rats and mice [71]. On this basis, animal studies must be carefully designed and carefully analyzed, or they cannot be interpreted with respect to human effects [72]. Here we will discuss the results of this systematic review, making clear distinguishment between animal model studies and human studies (Figure 3).

Increasing the number of injections did not produce additional loss in muscle strength and contractile material, as one might have suspected, suggesting that most of the muscle damage effects of BTX-A injection into muscles are caused by the first injection, or that the recovery period between injections was sufficient for partial recovery, thereby offsetting the potential damage induced by each injection.

Genetic alterations related to muscle atrophy/recovery through molecular biology were analyzed by five studies and showed how much impact a single BoNT-A injection can cause on a molecular basis. Mukund (2014) [49] realized that the direct action of BTX-A in skeletal muscle is relatively rapid, inducing dramatic transcriptional adaptation at one week and activating genes in competing pathways of repair and atrophy by gene-related impaired mitochondrial biogenesis.

Much like the findings of animal studies, human studies have also clearly shown atrophy in different muscle types after BTX-A injections. All six human studies that evaluated muscle histology showed atrophy, and when muscle recovery was assessed, there was no full recovery—Borodic (1992) [29] and Schroeder (2009) [41]. Bringing this idea into the context of facial aesthetics, the treatment of the Orbicularis oculi muscle, for example, with BTX-A sporadic injections could atrophy this muscle, but serial and controlled treatments could really maintain a certain degree of atrophy capable of allowing a smile with more open eyes, less caudal traction vector in the cranial part of this muscle postponing gravitational aging, and even give less contribution to the formation of the famous periorbital wrinkles, this time, as a secondary effect. Extrapolations of the powerful tool of muscle atrophy control through time using BTX-A injections could change completely the way BTX-A is used for aesthetic purposes. Dosages, injection intervals, and target muscles would be different from the patterns used nowadays. Instead of planning BTX-A injections to treat wrinkles, a modern anatomy understanding of the facial mimetic muscles as described by Boggio (2017) [74] would be of unparallel importance for aesthetic treatment planning [74]. New approaches for facial aesthetic treatments using BoNT-A could be completely based on mimetic facial muscle interactions and focused on reducing the activity of muscles that enhance gravitational aging (facial depressor muscles), such as the platysma muscle, for example, and preserving antigravitational muscles (elevator facial muscles), such as the frontalis (Figure 4).

After analyzing the results of this paper, we can attempt to answer the questions raised in the introduction (Table 16).

## 6. Conclusions

This systematic review showed evidence that seriated or single BoNT-A muscle injections can cause real muscle atrophy on a short or long-term basis, in animal models and in humans, in skeletal striated muscles of the limbs, facial masticatory muscles, and facial mimetic muscles. Theoretically, muscular architecture reprogramming is a possible new approach in aesthetics. Depressor facial muscles could be targeted to have some degree of atrophy with BoNT-A injections, while elevator facial muscles could be spared to some degree to maintain antigravitational traction forces and facilitate a lift effect.

## Figures and Tables

**Figure 1 toxins-14-00081-f001:**
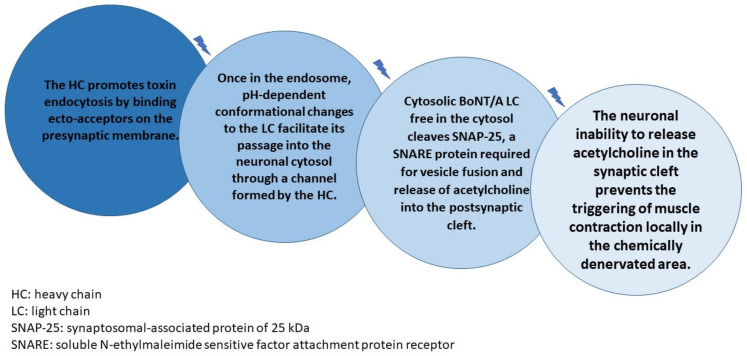
BoNT-A injection, the simplified mechanism of action cascade.

**Figure 2 toxins-14-00081-f002:**
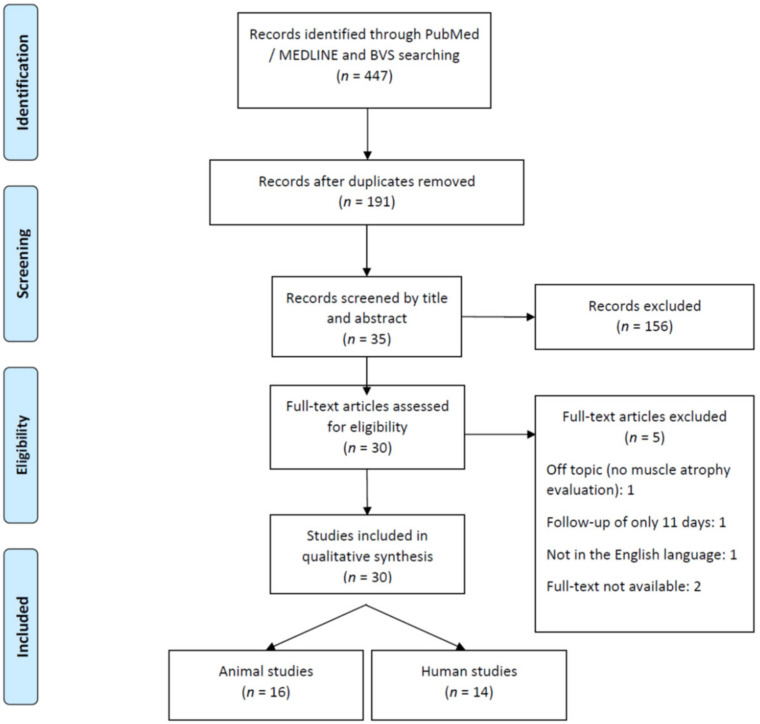
PRISMA—Flow Diagram of Article Selection for Review.

**Figure 3 toxins-14-00081-f003:**
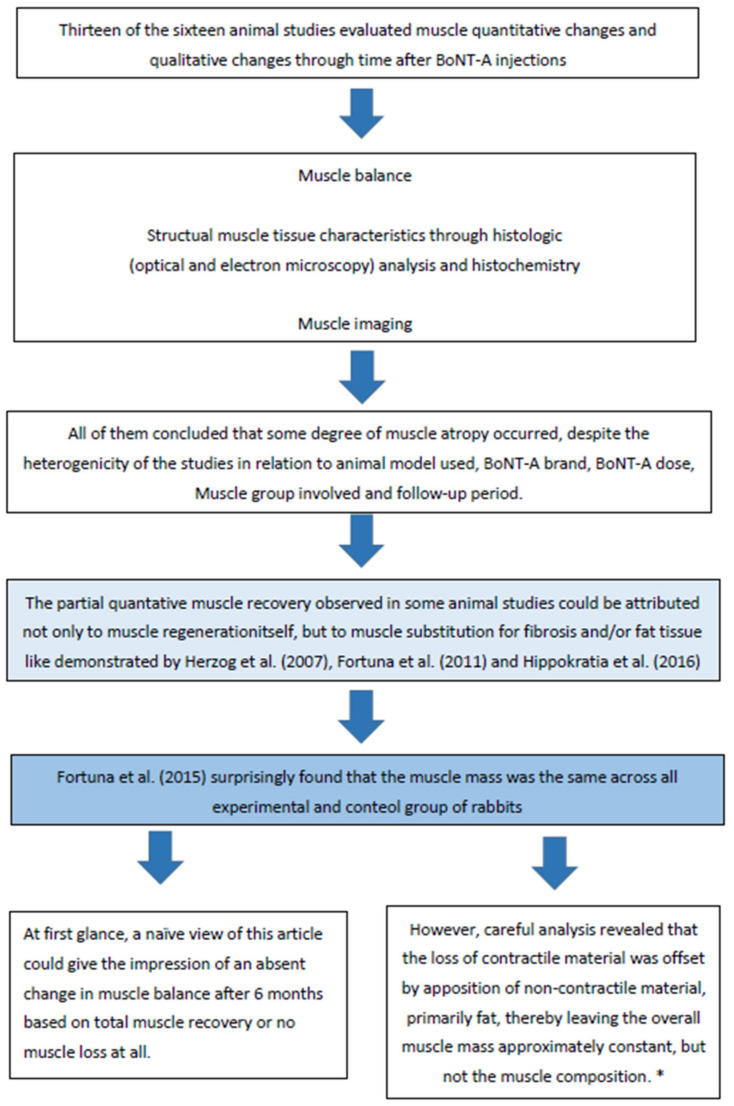
Animal model studies results—Discussion overview. * This finding might be of clinical relevance, because muscle volume measured using non-invasive imaging techniques (MRI, ultrasound) are sometimes used to approximate muscle mass in patient populations to determine progression of a disease or success of a treatment intervention—Damiano and Moreau (2008) [73]. Structural integrity and functional properties of muscles, rather than muscle mass or volume, might be more appropriate outcome measures to determine disease progression or aesthetics intervention effects.

**Figure 4 toxins-14-00081-f004:**
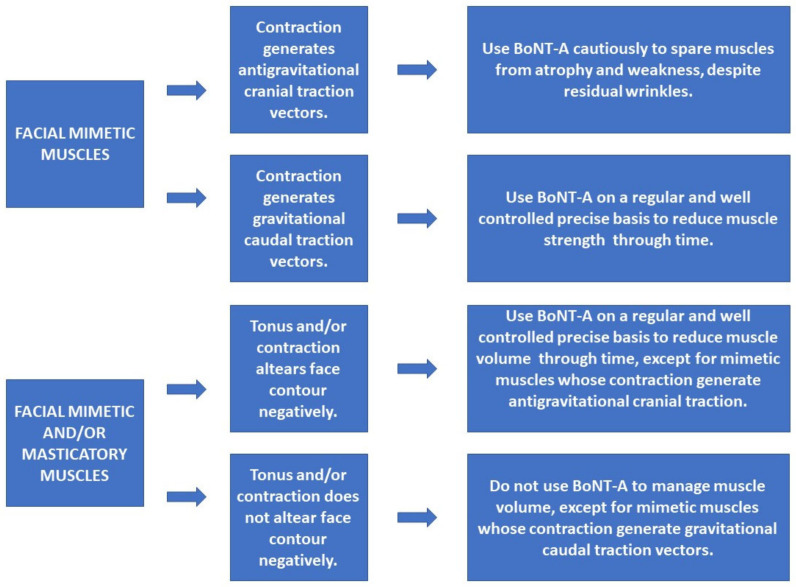
New approaches for facial aesthetic treatments using BoNT-A. The human imaging studies, similar to the animal studies, also show muscle atrophy and volume reduction. Koerte (2013) [47] showed a sustained atrophy and volume loss of approximately 50% in the procerus muscle. New perspectives on aesthetics BoNT-A treatments should consider not only facial mimetic muscles and their strength in relation to gravitational or antigravitational contraction vectors, but also their volume. Muscle volume control is also of aesthetic importance. The understanding that some degree of muscle volume reduction would bring positive aesthetic aspects for some mimetic muscles, such as the procerus and corrugators and some masticatory muscles such as the masseter, would also change the current BoNT-A injections patterns. On the other hand, some muscles should be spared from volume loss, such as the frontalis and the lateral aspect of the orbicularis oculi, to avoid facial skeletonization.

**Table 1 toxins-14-00081-t001:** Questions that should be answered, based on the evidence, after reading this paper.

Questions	Answers
Does the muscular impairment for contraction caused by BoNT-A really treats facial lines or causes muscle atrophy?	?
What is the relation of BoNT-A muscle injections and muscle atrophy in the long term?	?
Is it possible to modulate the level of muscle atrophy through time by using BoNT-A?	?
What if we used muscle atrophy caused by BoNT-A injections to optimize muscle architecture for facial aesthetic purposes?	?
What would it be like to reinterpret articles written in the last 30 years focused mainly on facial lines unveiling this concept of muscle atrophy? How many less subjective opportunities would arise? How classic BoNT-A injections techniques would be impacted?	?

**Table 2 toxins-14-00081-t002:** PubMed/MEDLINE and BVS (Biblioteca Virtual em Saúde) databases Search strategies.

Four Search Strategies Used, Initially:	
Search 1—PubMed/MEDLINE—((((BOTULINUM TOXIN TYPE A) OR (BOTULINUM A TOXIN)) OR (BOTULINUM NEUROTOXIN A)) OR (BOTOX)) AND (MUSCLE ATROPHY).	
Search 2—PubMed/MEDLINE—((((BOTULINUM TOXIN TYPE A) OR (BOTULINUM A TOXIN)) OR (BOTULINUM NEUROTOXIN A)) OR (BOTOX)) AND (MUSCULAR ATROPHY).	
Search 3—BVS—tw:((tw:(botulinum toxin type a)) OR (tw:(botulinum a toxin)) OR (tw:(botulinum neurotoxin a)) OR (tw:(botox)) AND (tw:(muscle atrophy))).	
Search 4—BVS—tw:((tw:(botulinum toxin type a)) OR (tw:(botulinum a toxin)) OR (tw:(botulinum neurotoxin a)) OR (tw:(botox)) AND (tw:(muscular atrophy))).	
To encompass all possible missing studies that could not be retrieved from Searches 1–4, the preferred MeSH term entries “Botulinum Toxin Type A” and “Muscular Atrophy” were matched with all their alternative MeSH term entries listed below:	
Botulinum toxin type A	Muscular atrophy
Clostridium Botulinum Toxin Type A	Atrophies, Muscular
Botulinum Toxin Type A	Atrophy, Muscular
Botulinum A Toxin	Muscular Atrophies
Toxin, Botulinum A	Atrophy, Muscle
Clostridium botulinum A Toxin	Atrophies, Muscle
Botulinum Neurotoxin A	Muscle Atrophies
Neurotoxin A, Botulinum	Muscle Atrophy
Meditoxin	Neurogenic Muscular Atrophy
Botox	Atrophies, Neurogenic Muscular
Neuronox	Atrophy, Neurogenic Muscular
Oculinum	Muscular Atrophies, Neurogenic
Vistabex	Muscular Atrophy, Neurogenic
OnabotulinumtoxinA	Neurogenic Muscular Atrophies
Onabotulinumtoxin A	Neurotrophic Muscular Atrophy
Vistabel	Atrophies, Neurotrophic Muscular
	Atrophy, Neurotrophic Muscular
	Muscular Atrophies, Neurotrophic
	Muscular Atrophy, Neurotrophic
	Neurotrophic Muscular Atrophies

All the 15 alternative MeSH term entries for “Botulinum Toxin Type A” and all the 19 alternative MeSH term entries for “Muscle Atrophy” listed above were individually added to Search 1, Search 2, Search 3, and Search 4, one at a time, to check if any other study would be retrieved. No other search limits were added.

**Table 3 toxins-14-00081-t003:** Inclusion and exclusion criteria.

A study was considered eligible for data extraction if it fulfilled the criteria bellow:
- Human or animal striated skeletal muscle atrophy analysis after botulinum toxin type A injection(s), and - Atrophy analyzed by imaging (ultrasonography (USG), nuclear magnetic resonance (NMR), computerized tomography (CT)), and/or by histological analysis and/or by biochemical analysis; and - Minimal follow-up of 3 months, and - The full manuscript was published in English.

**Table 4 toxins-14-00081-t004:** PICO-like structured reading of the eligible studies and data collection.

PICO-like structured reading of the eligible studies and data collection
Population/Problem (P) Intervention (I) Comparison group (C) Outcomes (O)
The following question was adopted to conduct data collection: “Are botulinum toxin type A injections (I) related to muscle atrophy (O) of animal or humans (P), when compared to not injected subjects or muscles (C)?”
Detailed data were collected in two different groups (animal and human) to fulfill comparative tables, including: presence of a control group, population number, population age, health condition, muscle systems analyzed, BoNT-A number of injections and dose, muscle atrophy confirmation or not, muscle atrophy identification tool and correlated changes, follow-up, and muscle atrophy recovery.

**Table 5 toxins-14-00081-t005:** Quality analysis form used in the systematic review.

Quality Analysis form Used in the Systematic Review.
Q1 Is there in the abstract an explanation of what was done and found? Q2 Is the scientific context clearly explained? Q3 Are the objectives clearly stated? Q4 Is the sampling size indicated? Q5 If yes, is the sampling size statistically justified? Q6 Are the characteristics of the subjects (height, weight, sex, healthy, or pathologic subject) described? Q7 What is the design of the study? (0: retrospective study; 1: case study; 2: prospective study). Q8 Is there a control group? (0: no, 1: contralateral member or nonrandomized control group, 2: randomized control group). Q9 How long is the follow up? (0: ≥3 and <6 months; 1: ≥6 months and <1 year; 2: ≥1 year) Q10 Is the reliability of the evaluation method clearly described? Q11 Are the results interpretable?Q12 Are the limitations of the study discussed?Q13 Is the conclusion clearly stated?

0: no description; 1: limited description; 2: good description.

**Table 6 toxins-14-00081-t006:** Quality assessment. ** maximum global score = 26.

	Q1	Q2	Q3	Q4	Q5	Q6	Q7	Q8	Q9	Q10	Q11	Q12	Q13	TOTAL **
Borodic (1992) [29]	1	2	2	2	0	1	2	1	2	1	1	0	2	17
Hamjian (1994) [30]	1	1	1	2	0	2	2	1	0	2	1	0	1	14
Ansved (1997) [31]	2	2	2	2	0	2	1	1	2	1	1	0	1	17
Fanucci (2001) [32]	2	2	2	2	0	2	2	1	0	2	2	0	2	19
To (2001) [33]	2	2	2	2	0	2	2	1	2	1	2	0	2	20
Kim (2005) [34]	2	2	1	2	0	2	2	0	2	1	2	0	2	18
Shen (2006) [35]	2	2	2	2	0	2	2	2	1	2	2	0	2	21
Singer (2006) [36]	2	2	2	2	0	2	2	0	1	1	2	2	2	20
Herzog (2007) [37]	2	2	2	2	0	2	2	2	1	2	2	2	2	23
Frick (2007) [38]	2	2	1	2	0	2	2	2	0	1	2	1	2	19
Kwon (2007) [39]	2	2	2	2	0	2	2	2	1	2	2	1	2	22
Lee (2007) [40]	2	2	2	2	0	2	2	0	2	1	2	0	2	19
Schroeder (2009) [41]	2	2	2	2	0	1	2	1	2	1	2	0	2	19
Babuccu (2009) [42]	2	2	2	2	0	2	2	2	0	2	2	0	2	20
Tsai (2010) [43]	2	2	1	2	0	1	2	1	2	1	1	1	2	18
Fortuna (2011) [44]	2	2	2	2	0	2	2	2	1	1	1	1	2	20
Fortuna (2013a) [45]	2	2	2	2	0	1	2	2	1	1	2	1	2	20
Van Campenhout (2013) [46]	2	2	2	2	0	1	2	0	1	2	2	2	2	20
Koerte (2013) [47]	2	2	2	2	0	2	2	1	2	1	2	0	1	19
Fortuna (2013b) [48]	2	2	2	2	0	2	2	2	2	1	2	2	1	22
Mukund (2014) [49]	1	2	2	2	0	2	2	1	2	2	2	1	2	21
Fortuna (2015) [50]	2	2	2	2	0	2	2	2	2	2	2	0	2	22
Caron (2015) [51]	2	2	2	2	0	2	2	2	2	1	2	0	1	20
Valentine (2016) [52]	2	2	2	2	0	2	2	1	2	1	2	1	1	20
Li (2016) [53]	1	1	2	2	0	2	1	0	2	0	1	0	1	13
Kocaelli (2016) [54]	2	2	2	2	0	2	2	2	0	2	2	1	2	21
Hart (2017) [55]	2	2	2	2	0	2	2	2	2	2	1	2	1	22
Han (2018) [56]	2	2	2	2	0	2	2	0	1	1	2	1	1	18
Alexander (2018) [57]	2	2	2	2	0	2	2	1	1	2	2	2	2	22
Lima (2018) [58]	2	2	2	2	0	2	2	2	0	1	2	0	2	19

**Table 7 toxins-14-00081-t007:** Systematic review—Summary table of the results (PART 1).

Author (Year)	Human/Animal	Control Group	Age	Population (Number)	Health Condition
Borodic (1992) [29]	Human	Yes	56–91 years	14	Blepharospasm/Meige’s disease
Hamjian (1994) [30]	Human	Contralateral muscle	25–49 years	10	Healthy
Ansved (1997) [31]	Human	Yes	32–54 years	22	Cervical dystonia
Fanucci (2001) [32]	Human	Contraleteral Muscle	29–54 years	30	Piriformis muscle syndrome (PMS)
To (2001) [33]	Human	Yes	16–32 years	15	Masseteric muscle hypertrophy
Kim (2005) [34]	Human	No	Teenagers—40s	383	Masseteric muscle hypertrophy
Shen (2006) [35]	Animal (Sprague-Dawley rats)	Yes	1 month	56	Healthy
Singer (2006) [36]	Human	No	16–40 years	8	Chronic anterior knee pain and related disability
Herzog (2007) [37]	Animal (New Zealand white rabbits)	Yes	1 year	25	Healthy
Frick (2007) [38]	Animal (Sprague-Dawley rats)	Contralateral muscle	Mature	39	Healthy
Kwon (2007) [39]	Animal (New Zealand rabbits)	Yes	4 weeks	21	Healthy
Lee (2007) [40]	Human	No	20–29 years	10	Healthy (square face)
Schroeder (2009) [41]	Human	Contralateral muscle	31–47 years	2	Healthy
Babuccu (2009) [42]	Animal (Wistar rats)	Yes	15-day-old	49	Healthy
Tsai (2010) [43]	CD^®^ (SD) IGS rats	Contralateral muscle	Mature	60	Healthy
Fortuna (2011) [44]	Animal (New Zealand White rabbits)	Yes	1 year	20	Healthy
Fortuna (2013a) [45]	Animal (New Zealand White rabbits)	Yes	Mature	17	Healthy
Van Campenhout (2013) [46]	Human	No	Children	7	Cerebral palsy (symmetric spastic diplegia)
Koerte (2013) [47]	Human	Yes	34–50 years	4	Healthy
Fortuna (2013b) [48]	Animal (New Zealand White rabbits)	Yes	1 year	27	Healthy
Mukund (2014) [49]	Animal (Harlan Sprague-Dawley rats)	Contralateral muscle	3 months	20	Healthy
Fortuna (2015) [50]	Animal (New Zealand White rabbits)	Yes	1 year	23	Healthy
Caron (2015) [51]	Animal (Sprague-Dawley rats)	Yes	Mature	27	Healthy
Valentine (2016) [52]	Human	Different muscle same participant	6–16 years	10	Cerebral palsy
Li (2016) [53]	Human	No	40–59 years	3	Strabismus
Kocaelli (2016) [54]	Animal (Sprague-Dawley rats)	Yes	5–6 months	30	Healthy
Hart (2017) [55]	Animal (New Zealand White rabbits)	Yes	1 year	22	Healthy
Han (2018) [56]	Animal (Cynomolgus monkey—*Macaca fascicularis*)	No	9 years	1	Healthy
Alexander (2018) [57]	Human	Baseline status same participant	5–13 years	11	Cerebral palsy
Lima (2018) [58]	Animal (Wistar rats)	Yes	10-week-old	50	Healthy

Systematic review—Summary table of the results (PART 1). Human studies 
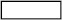
 Animal studies 
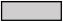
.

**Table 8 toxins-14-00081-t008:** Systematic review—Summary table of the results (PART 2).

Author (Year)	BoNT-A Number of Injections and Dose	Follow-Up
Borodic (1992) [29]	2–19 injections over 1–5.5 years. Dose?	1–52 weeks
Hamjian (1994) [30]	1 injection. Dose 10 units of BoNT-A (Oculinum^®^) ^#^	0–100 days
Ansved (1997) [31]	Number? 2–4 years of treatment. Mean cumulative dose 2.815 units of BoNT-A	2–4 years
Fanucci (2001) [32]	1 or 2 injections. Dose 200 units of BoNT-A (Botox^®^) ^##^	0–3 months
To (2001) [33]	1 or 2 injections. Dose 100–300 units of BoNT-A (Dysport^®^) ^###^ per side	0–1 year
Kim (2005) [34]	1 or 2 injections. Dose 100–140 units of BoNT-A (Dysport^®^) ^###^ per side	0–2 years
Shen (2006) [35]	1 injection. Dose 6 units/kg body weight of BoNT-A (Botox^®^) ^##^	0–360 days
Singer (2006) [36]	1 injection. Dose 300–500 units of BoNT-A (Dysport^®^) ^###^	0–24 weeks
Herzog (2007) [37]	1–6 injetions over 6 months. Dose 3,5 units/kg body weight of BoNT-A (Botox^®^) ^####^ per injetion	1–6 months
Frick (2007) [38]	1 injection. Dose 0.625 units or 2.5 units or 10 units/kg body weight of BoNT-A (Botox^®^) ^##^	128 days
Kwon (2007) [39]	1 injection. Dose 5–15 units of BoNT-A	4–24 weeks
Lee (2007) [40]	1 injection. Dose 25 units of BoNT-A (Botox^®^) ^##^	0–12 months
Schroeder (2009) [41]	1 injection. Dose 75 units of BoNT-A (Xeomin^®^) ^#####^	3–12 months
Babuccu (2009) [42]	1 injection. Dose 0.4 units BoNT-A (Botox^®^) ^######^ per muscle	4 months
Tsai (2010) [43]	1 or 2 injetions. Dose 2.5 ng of BoNT-A (Botox^®^) ^##^ per side (single injection group) or (two injections group full dose—30 weeks apart) or 1.25 ng (two injections group half dose—30 weeks apart)	1–58 weeks
Fortuna (2011) [44]	1 or 3 or 6 monthly injections. Dose 3.5 units/Kg of BoNT-A (Botox^®^) ^####^ per muscle group, per side, per month	1–6 months
Fortuna (2013a) [45]	6 monthly injections. Dose 3.5 units/Kg of BoNT-A (Botox^®^) ^####^ per muscle group, per side, per month	6 months
Van Campenhout (2013) [46]	1 injection. Dose 2 units/Kg/psoas muscle of BoNT-A (Botox^®^) ^##^	0–6 months
Koerte (2013) [47]	1 injection. Dose 20 units of BoNT-A (Botox^®^) ^##^	0–12 months
Fortuna (2013b) [48]	6 monthly injections. Dose 3.5 units/Kg of BoNT-A (Botox^®^) ^####^ per muscle group, per side, per month	6–12 months
Mukund (2014) [49]	1 injection. Dose 6 units/Kg of BoNT-A (Botox^®^) ^##^ per side	1–52 weeks
Fortuna (2015) [50]	1, 2, or 3 injections (every 3 months). Dose 3.5 units/Kg of BoNT-A (Botox^®^) ^####^ per muscle group, per side, per injection	6–12 months
Caron (2015) [51]	1 injection. Dose 15 units/Kg of BoNT-A (Dysport^®^) ^#######^ per side	12–400 days
Valentine (2016) [52]	1–15 injections. Dose 2–6 units/Kg of BoNT-A (Botox^®^) ^##^ per side	3.5 months–3 years
Li (2016) [53]	1–2 injections. Dose 3.75–7.5 units of BoNT-A (Botox^®^) ^##^ per side	6–18 months
Kocaelli (2016) [54]	1 injection. Dose 0.5 units of BoNT-A (Botox^®^) ^##^ per muscle, per side	12 weeks
Hart (2017) [55]	1, 2, or 3 injections (every 3 months). Dose 3.5 units/Kg of BoNT-A (Botox^®^) ^####^ per muscle group, unilateral, per injection	6–12 months
Han (2018) [56]	10 (one injection every two weeks). Dose 2 units/Kg of BoNT-A (Nabota^®^) ^########^	0–21 weeks
Alexander (2018) [57]	1 injection. Dose 1.4–4.8 units/Kg of BoNT-A (Botox^®^) ^##^ per side	0–25 weeks
Lima (2018) [58]	1 injection. Dose 5 units of BoNT-A (Dysport^®^) ^###^ per side	12 weeks

Human studies 
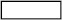
 Animal studies 
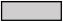
 # (Oculinum^®^)—Allergan Corp., Irvine, CA. ## (Botox^®^)—Allergan Corp., Irvine, CA. ### (Dysport^®^) Ipsen Ltd., Slough, United Kingdom. #### (Botox^®^) Allergan Inc., Toronto, Ont., Canada. ##### (Xeomin^®^) Merz Pharma, Germany. ###### (Botox^®^) Allergan Pharmaceuticals, Ireland. ####### (Dysport^®^) Ipsen Ltd., Boulogne-Billancourt, France. ######## (Nabota^®^) Daewoong Pharmaceutical Hwaseong, Korea.

**Table 9 toxins-14-00081-t009:** Animal studies—Muscle balance.

Muscle Atrophy Identification Tool	Herzog (2007) [37] Quadriceps Femoris 25 New Zealand White Rabbits	Frick (2007) [38] Tibialis 39 Sprague-Dawley Rats	Babuccu (2009) [42] Masseter and Temporalis 49 Wistar Rats	Tsai (2010) [43] Gastrocnemius 60 CD^®^ (SD) IGS Rats	Fortuna (2011) [44] Quadriceps Femoris 20 New Zealand White Rabbits	Fortuna (2013a) [45] Quadriceps Femoris 20 New Zealand White Rabbits	Fortuna (2013b) [48] Quadriceps Femoris 27 New Zealand White Rabbits	Fortuna (2015) [50] Quadriceps Femoris 23 New Zealand White Rabbits	Caron (2015) [51] Gastrocnemius 27 Sprague-Dawley Rats	Lima (2018) [58] Gastrocnemius 50 Wistar Rats
Balance (immediately post-sacrifice muscle harvest or muscle harvest under general anesthesia)	Wet muscle mass	Wet muscle mass	Wet muscle mass	Wet muscle mass	Wet muscle mass	Wet muscle mass	Wet muscle mass	Wet muscle mass	Wet muscle mass and muscle weight/body weight ratio	Wet muscle mass
	Mean percent loss of muscle mass of 36% at 1 month and 49% at 6 months.	Significant (*p* < 0.05) decrease of 20% in (group 0.625 units), 33.4% in (group 2.5 units) and 50% in (group 10 units) at day 128. No recovery at day 128.	Significantly diminished *p* = 0.0001 (masseter) and *p* = 0.001 (temporalis). No recovery.	Reduction of 10.7% (±3.8) at 58 weeks after a single BoNT-A injection, 29.7% (±8.2) after repeated injections half dose and a reduction of 41.7% (±6.1) at 58 weeks after repeated injections of full dose. Partial recovery at 58 weeks.	Significant atrophy (*p* < 0.0001). Mean quadriceps femoris muscle mass reduction of 45% (1 month group), 60% (3 months group), and 56% (6 months group). No recovery.	Significant atrophy (*p* < 0.001). Mean quadriceps femoris muscle mass reduction of 52%. No recovery.	Reduction of 52% (*p* < 0.001) at 6 months of repeated monthly BoNT-A injections and a sustained reduction of 18% (*p* < 0.001) at 6 months after the last BoNT-A injection. Partial after 6 months of the last BoNT-A injection.	No alteration at 6 months after the last BoNT-A injection (*p* > 0.05).	No alteration at 6 months. Significantly lower weight (*p* < 0.001) at 12 days post BoNT-A injection. Significantly lower weight (*p* < 0.001) at 128.43 ± 7.43 days post BoNT-A injection. Significantly partial weight recovery (*p* < 0.001) at 371.83 ± 24.82 days post BoNT-A injection. No recovery.	Significant reduction of 37% (*p* < 0.001).

**Table 10 toxins-14-00081-t010:** Animal studies—Hystologic (optical and electron microscopy) analysis and histochemistry.

Muscle Atrophy Identification Tool	Herzog (2007) [37] Quadriceps Femoris 25 New Zealand White Rabbits	Frick (2007) [38] Tibialis 39 Sprague-Dawley Rats	Babuccu (2009) [42] Masseter and Temporalis 49 Wistar Rats	Tsai (2010) [43] Gastrocnemius 60 CD^®^ (SD) IGS Rats	Fortuna (2011) [44] Quadriceps Femoris 20 New Zealand White Rabbits	Fortuna (2013a) [45] Quadriceps Femoris 20 New Zealand White Rabbits	Fortuna (2013b) [48] Quadriceps Femoris 27 New Zealand White Rabbits	Fortuna (2015) [50] Quadriceps Femoris 23 New Zealand White Rabbits	Kocaelli (2016) [54] Masseter and Gluteal 30 Sprague-Dawley Rats
Histologic analysis (optical microscopy)/ histochemistry	Muscle structure (qualitative)	Muscle structure (qualitative)	Muscle structure (qualitative)	Muscle structure (qualitative)	Muscle structure (qualitative)	Muscle structure (qualitative)	Muscle structure (qualitative)	Muscle structure (qualitative)	Muscle structure (qualitative)
	Replacement of contractile fibers with fat.				Fatty infiltration at 3 and 6 months (increased). No recovery.				Increase in the collagen fibers forming perimysium around the striated muscle cells at 12 weeks.
	Muscle structure (percentage of contractile material)	Muscle structure (percentage of contractile material)	Muscle structure (percentage of contractile material)	Muscle structure (percentage of contractile material)	Muscle structure (percentage of contractile material)	Muscle structure (percentage of contractile material)	Muscle structure (percentage of contractile material)	Muscle structure (percentage of contractile material)	Muscle structure (percentage of contractile material)
		Significant (*p* < 0.05) decrease at day 128. No recovery at day 128.			Significantly reduced (*p* < 0.05) (6 months group) for 43% (±9.7) vastus lateralis, for 70% (±8.0) rectus femoris, for 78% (±4.2) vastus medialis. No recovery.	Reduction of 36.1% (±16.9), (*p* < 0.001). No recovery.	Reduction of 36.1% (±16.9), (*p* < 0.001) at 6 months of repeated monthly BoNT-A injections and a sustained reduction of 22.2% (±2.0) at 6 months after the last BoNT-A injection. Partial recovery at 6 months.	Reduction of 40.8% (±6.0), at 6 months after 1 BoNT-A injection, reduction of 37.5% (±6.1), at 6 months after 2 BoNT-A injection, reduction of 40.1% (±11.8), at 6 months after 3 BoNT-A injection. No recovery.	
	Muscle structure (atrophy scoring/quantitative analysis)	Muscle structure (atrophy scoring/quantitative analysis)	Muscle structure (atrophy scoring/quantitative analysis)	Muscle structure (atrophy scoring/quantitative analysis)	Muscle structure (atrophy scoring/quantitative analysis)	Muscle structure (atrophy scoring/quantitative analysis)	Muscle structure (atrophy scoring/quantitative analysis)	Muscle structure (atrophy scoring/quantitative analysis)	Muscle structure (atrophy scoring/quantitative analysis)
			Stratification degree of the muscle, nucleus internalization, multinucleation, myofibril diameter, and myonecrosis compatible with muscle atrophy. No recovery at 4 months.						Significant (*p* < 0.001) decrease of diameters of muscle fibers in bundles and fascicles at 12 weeks.
Histologic analysis (electron microscopy)/ histochemistry	Muscle ultrastructure	Muscle ultrastructure	Muscle ultrastructure	Muscle ultrastructure	Muscle ultrastructure	Muscle ultrastructure	Muscle ultrastructure	Muscle ultrastructure	Muscle ultrastructure
				Sarcomere distorsion (mild distruction at 8 weeks). Partial recovery at 26 weeks.					Myofibrils atrophic changes characterized by: decrease in myofibrillar diameters and myofibrillolysis, dilatations in the terminal cisternae and T-tubules, disorganized Z bands, vacuolar appearance as a result of dilatation in the sarcoplasmic reticulum cisternae and mitochondrial swelling.

**Table 11 toxins-14-00081-t011:** Animal studies—Imaging.

Muscle Atrophy Identification Tool	Kwon (2007) [39] Masseter 21 New Zealand Rabbits	Han (2018) [56] Paraspinal 01 Cynomolgus Monkey—Macaca Fascicularis
Magnetic resonance imaging (MRI)	Muscle cross-sectional areas at T12–L1, L1–L2, L2–L3, L3–L4 and L4–L5 levels	Muscle cross-sectional areas at T12–L1, L1–L2, L2–L3, L3–L4, and L4–L5 levels
		Significant atrophy with decreased cross-sectional areas by 4%, 2%, 8%, 12%, and 8%, respectively, at 21 weeks (the peak was at 11 weeks). Partial recovery at 21 weeks.
Computed tomography (CT) scan	Muscle volume	Muscle volume
	Reduction of 19.72% (±4.80) in Group 2 and of 21.34% (±5.37) in Group 3 at 8 weeks. Reduction of 13.76% (±5.34) in Group 2 and of 18.41% (±3.15) in Group 3 at 24 weeks. Partial recovery at 24 weeks.	

**Table 12 toxins-14-00081-t012:** Animal studies—Direct and indirect muscle atrophy identification via molecular biology.

Molecular Biology Alterations	Articles
Upregulation of proapoptotic: anti-apoptotic protein ratio ((Bax:Bcl-2)ratio) significantly had an 83.3 fold increase, peak at 4 weeks. *p* < 0.01	Tsai (2010) [43].
Muscle substitution for adipose tissue determined by adipocyte-related molecules upregulation of adiponectin (APN), Leptin, adipocyte binding protein 2 (AP2), and adipogenic lineage marker upregulation of peroxisome proliferator-activated receptor γ (PPARγ). The APN, Leptin, AP2, and PPARγ were significantly upregulated after BoNT-A injections. *p* < 0.05	Hart (2017) [55].
Muscle atrophy inferred via molecular biology in regard to upregulation of Transforming Growth Factor-beta TGF-β; upregulation of Nuclear Factor-kappaB (NF-κB); upregulation of p53/Cell cycle control; upregulation of Inhibitor of DNA binding (ID) proteins—Id1, Id2, Id3, Id4, and muscle RING-finger protein-1 (MuRF1) upregulation.	Mukund (2014) [49]. Fortuna (2015) [50].
Muscle atrophy and muscle atrophy recovery response indirectly identified via NMJ restoration (muscle-specific receptor tyrosine kinase (MuSK) upregulation, nicotinic acetylcholine receptor (nAChR) upregulation), protection against muscle cell apoptosis (P21 protein upregulation), myogenesis modulation/muscle regeneration (insulin-like growth factor-1 (IGF-1) upregulation, myogenin upregulation, and mitogen-activated protein kinase (MAPK) upregulation).	Shen (2006) [35]. Mukund (2014) [49]. Fortuna (2015) [50].

**Table 13 toxins-14-00081-t013:** Animal studies—Molecular biology.

Muscle Atrophy Identification Tool	Shen (2006) [35] Gastrocnemius 56 Sprague-Dawley Rats	Tsai (2010) [43] Gastrocnemius 60 CD^®^ (SD) IGS Rats	Mukund (2014) [49] Tibialis Anterior 20 Sprague-Dawley Rats	Fortuna (2015) [50] Quadriceps Femoris 23 New Zealand White Rabbits	Hart (2017) [55] Quadriceps Femoris 22 New Zealand White Rabbits
Molecular biology (Real-Time Quantitative Polymerase Chain Reaction (qPCR), and/or Microarray Data Analysis, and/or Western blot analysis)	Indirect atrophy identification via upregulation of gene and molecule expression signaling neuromuscular junction (NMJ) restoration, protection against muscle cell apoptosis, myogenesis modulation/muscle regeneration.				
	NMJ restoration	NMJ restoration	NMJ restoration	NMJ restoration	NMJ restoration
	Muscle-specific receptor tyrosine kinase (MuSK) significant upregulation (*p* < 0.05) from day 3 to day 60 Nicotinic acetylcholine receptor (nAChR) significant upregulation (*p* < 0.05) from day 3 to day 14				
	Protection against muscle cell apoptosis	Protection against muscle cell apoptosis	Protection against muscle cell apoptosis	Protection against muscle cell apoptosis	Protection against muscle cell apoptosis
	P21 protein significant (*p* < 0.05) upregulation from day 3 to day 30				
	Myogenesis modulation/muscle regeneration	Myogenesis modulation/muscle regeneration	Myogenesis modulation/muscle regeneration	Myogenesis modulation/muscle regeneration	Myogenesis modulation/muscle regeneration
	Insulin-like growth factor-1 (IGF-1) significant upregulation (*p* < 0.05) from day 3 to day 60 Myogenin significant upregulation (*p* < 0.05) from day 3 to day 90				
	Myogenesis modulation/muscle regeneration	Myogenesis modulation/muscle regeneration	Myogenesis modulation/muscle regeneration	Myogenesis modulation/muscle regeneration	Myogenesis modulation/muscle regeneration
				Insulin-like growth factor-1 (IGF-1) significant upregulation (*p* < 0.05) (at 6 months) Recovery not evaluated	
	Direct atrophy identification via upregulation of proapoptotic: anti-apoptotic protein ratio (Bax:Bcl-2)	Direct atrophy identification via upregulation of proapoptotic: anti-apoptotic protein ratio (Bax:Bcl-2)	Direct atrophy identification via upregulation of proapoptotic: anti-apoptotic protein ratio (Bax:Bcl-2)	Direct atrophy identification via upregulation of proapoptotic: anti-apoptotic protein ratio (Bax:Bcl-2)	Direct atrophy identification via upregulation of proapoptotic: anti-apoptotic protein ratio (Bax:Bcl-2)
		Ratio significantly 83.3 fold increase (*p* < 0.01) (peak at 4 weeks) Recovery at 8 weeks			
	Direct atrophy identification via upregulation of Transforming Growth Factor-beta TGF-β	Direct atrophy identification via upregulation of Transforming Growth Factor-beta TGF-β	Direct atrophy identification via upregulation of Transforming Growth Factor-beta TGF-β	Direct atrophy identification via upregulation of Transforming Growth Factor-beta TGF-β	Direct atrophy identification via upregulation of Transforming Growth Factor-beta TGF-β
				TGF-β significantly upregulated (*p* < 0.05) (at 6 months) Recovery not evaluated	
	Direct atrophy identification via muscle RING-finger protein-1 (MuRF1)	Direct atrophy identification via muscle RING-finger protein-1 (MuRF1)	Direct atrophy identification via muscle RING-finger protein-1 (MuRF1)	Direct atrophy identification via muscle RING-finger protein-1 (MuRF1)	Direct atrophy identification via muscle RING-finger protein-1 (MuRF1)
				MuRF1 significantly upregulated (*p* < 0.05) (at 6 months) Recovery not evaluated	
	Direct atrophy identification via muscle substitution for adipose tissue. Adipocyte-related molecules upregulation of adiponectin (APN), Leptin, adipocyte binding protein 2 (AP2), and adipogenic lineage marker upregulation of peroxisome proliferator-activated receptor γ (PPARγ)	Direct atrophy identification via muscle substitution for adipose tissue. Adipocyte-related molecules upregulation of adiponectin (APN), Leptin, adipocyte binding protein 2 (AP2), and adipogenic lineage marker upregulation of peroxisome proliferator-activated receptor γ (PPARγ)	Direct atrophy identification via muscle substitution for adipose tissue. Adipocyte-related molecules upregulation of adiponectin (APN), Leptin, adipocyte binding protein 2 (AP2), and adipogenic lineage marker upregulation of peroxisome proliferator-activated receptor γ (PPARγ)	Direct atrophy identification via muscle substitution for adipose tissue. Adipocyte-related molecules upregulation of adiponectin (APN), Leptin, adipocyte binding protein 2 (AP2), and adipogenic lineage marker upregulation of peroxisome proliferator-activated receptor γ (PPARγ)	Direct atrophy identification via muscle substitution for adipose tissue. Adipocyte-related molecules upregulation of adiponectin (APN), Leptin, adipocyte binding protein 2 (AP2), and adipogenic lineage marker upregulation of peroxisome proliferator-activated receptor γ (PPARγ)
					APN, Leptin, AP2, and PPARγ significantly upregulated (*p* < 0.05) (at 6 months after 3 BoNT-A injections every 3 months, except for Leptin, which had partial recovery after 3 BoNT-A injections)

**Table 14 toxins-14-00081-t014:** Human studies—Histologic (optical and electron microscopy) analysis and histochemistry.

Muscle Atrophy Identification Tool	Borodic (1992) [29] Orbicularis Oculi 14	Ansved (1997) [31] Vastus Lateralis (Non-Target Muscle) 22	Kim (2005) [34] Masseter 383	Schroeder (2009) [41] Gastrocnemius 2	Valentine (2016) [52] Gastrocnemius 10	Li (2016) [53] Medial Rectus (Extraocular Muscle) 3
Histologic analysis (optical microscopy)/ histochemistry	Morphometric measurements of muscle fibers	Morphometric measurements of muscle fibers	Morphometric measurements of muscle fibers	Morphometric measurements of muscle fibers	Morphometric measurements of muscle fibers	Morphometric measurements of muscle fibers
	Reduced and irregular diameter at 3 months (*p* < 0.05). Partial recovery at 6 months.	Mean diameter reduction of type IIB fibers of 19.6% after 2–4 years of BoNT-A treatement, (*p* < 0.05).				
	Muscle structure	Muscle structure	Muscle structure	Muscle structure	Muscle structure	Muscle structure
			Muscle atrophy, necrosis, and hyaline degeneration at 4 months.	Muscle atrophy and Mild increase of the number of perimysial fat cells. Muscle fiber area reduction of 24% at 12 months. Partial recovery at 12 months.	Muscle atrophy.	Fibrosis with no identifiable muscle fibers.
Histologic analysis (electron microscopy)/ histochemistry	Muscle ultrastructure	Muscle ultrastructure	Muscle ultrastructure	Muscle ultrastructure	Muscle ultrastructure	Muscle ultrastructure
				Muscle atrophy of a considerable number of muscle fibers at 12 months. Partial recovery at 12 months.	Atrophic muscle fibers, Myofibrillar disorganization, redundant basal lamina, cores, and wrinkling of the sarcolemmal membrane.	

**Table 15 toxins-14-00081-t015:** Human studies—Imaging.

Muscle Atrophy Identification Tool	Hamjian (1994) [30] Extensor Digitorum 10	Fanucci (2001) [32] Piriformis 30	To (2001) [33] Masseter 15	Kim (2005) [34] Masseter 383	Singer (2006) [36] Vastus Lateralis 8	Lee (2007) [40] Masseter 10	Schroeder (2009) [41] Gastrocnemius 2	Van Campenhout (2013) [46] Psoas 7	Koerte (2013) [47] Procerus 4	Alexander (2018) [57] Gastrocnemius 11
Ultrasound	Muscle thickness	Muscle thickness	Muscle thickness	Muscle thickness	Muscle thickness	Muscle thickness	Muscle thickness	Muscle thickness	Muscle thickness	Muscle thickness
	Decrease of 16% at peak (day 42), (*p* < 0.03). Recovery (Partial? Complete?) 100 days		Median decrease of 30.9% at peak (3 months) and 13.4% (1 year), (*p* < 0.001). Partial recovery 1 year.	Average decrease of 31% (3 months after BoNT-A injection), (*p* not calculated). Partial recovery 2 years.						
	Muscle Volume	Muscle Volume	Muscle Volume	Muscle Volume	Muscle Volume	Muscle Volume	Muscle Volume	Muscle Volume	Muscle Volume	Muscle Volume
	Decrease of 40% at peak (day 42), (*p* < 0.03). Recovery (Partial? Complete?) 100 days.									
Magnetic resonance imaging (MRI)	T2 short tau inversion recovery (S-TIR) weighted sequence	T2 short tau inversion recovery (S-TIR) weighted sequence	T2 short tau inversion recovery (S-TIR) weighted sequence	T2 short tau inversion recovery (S-TIR) weighted sequence	T2 short tau inversion recovery (S-TIR) weighted sequence	T2 short tau inversion recovery (S-TIR) weighted sequence	T2 short tau inversion recovery (S-TIR) weighted sequence	T2 short tau inversion recovery (S-TIR) weighted sequence	T2 short tau inversion recovery (S-TIR) weighted sequence	T2 short tau inversion recovery (S-TIR) weighted sequence
		Muscular atrophy at 3 months.								
	Signal Intensity (S.I.)	Signal Intensity (S.I.)	Signal Intensity (S.I.)	Signal Intensity (S.I.)	Signal Intensity (S.I.)	Signal Intensity (S.I.)	Signal Intensity (S.I.)	Signal Intensity (S.I.)	Signal Intensity (S.I.)	Signal Intensity (S.I.)
		High intensity (compatible with muscle atrophy) at 3 months.					High intensity (compatible with muscle atrophy) at 12 months.			
	Muscle cross-sectional area	Muscle cross-sectional area	Muscle cross-sectional area	Muscle cross-sectional area	Muscle cross-sectional area	Muscle cross-sectional area	Muscle cross-sectional area	Muscle cross-sectional area	Muscle cross-sectional area	Muscle cross-sectional area
							Reduction of 14–19% at 3 months, of 27% at 6 months (peak), and 12–22% at 12 months, (*p* not calculated). Partial recovery at 12 months.			
	Muscle volume	Muscle volume	Muscle volume	Muscle volume	Muscle volume	Muscle volume	Muscle volume	Muscle volume	Muscle volume	Muscle volume
								Reduction of 20% at 2 months and sustained at 6 months, (*p* = 0.004). No recovery at 6 months.	Reduction of 46% to 48% at 1 month and sustained at 12 months, (*p* not calculated). No recovery at 12 months.	Reduction of 5.9% at 4 weeks, of 9.4% at 13 weeks (peak reduction), of 6.8% at 25 weeks, (*p* < 0.05). Partial recovery from 13 to 25 weeks.
Computed tomography (CT) scan	Muscle cross-sectional area	Muscle cross-sectional area	Muscle cross-sectional area	Muscle cross-sectional area	Muscle cross-sectional area	Muscle cross-sectional area	Muscle cross-sectional area	Muscle cross-sectional area	Muscle cross-sectional area	Muscle cross-sectional area
					Mean decrease of 12.4% (+5%) at 12 weeks (*p* < 0.05).					
Cephalometry	Soft-tissue bigonial distance	Soft-tissue bigonial distance	Soft-tissue bigonial distance	Soft-tissue bigonial distance	Soft-tissue bigonial distance	Soft-tissue bigonial distance	Soft-tissue bigonial distance	Soft-tissue bigonial distance	Soft-tissue bigonial distance	Soft-tissue bigonial distance
						Decrease from 131 mm (±4.9) to 123.5 mm (±3.0) at 3 months (peak), (*p* < 0.05) from months 1 to 7, and sustained decrease to 130.1 mm (±4.6) at 12 months.				

**Table 16 toxins-14-00081-t016:** Possible and plausible evidence-based answers for the questions raised in the introduction.

Questions	Answers
Does the muscular impairment for contraction caused by BoNT-A really treat facial lines or cause muscle atrophy?	Muscle atrophy occurs after BoNT-A injections. Facial lines are, only in part, treated by BoNT-A injections.
What is the relationship betweenf BoNT-A muscle injections and muscle atrophy in the long term?	Muscles tend to maintain atrophy or have partially recover after BoNT-A injections.
Is it possible to modulate the level of muscle atrophy through time by using BoNT-A?	At least theoretically it is, and further studies could help us master this new frontier in facial aesthetics.
What if we used muscle atrophy caused by BoNT-A injections to optimize muscle architecture for facial aesthetic purposes?	It seems smart to use the atrophy after BoNT-A injections as a tool for aesthetic purposes instead of the old idea of an adverse event.
What would it be like to reinterpret articles written in the last 30 years that focused mainly on facial lines unveiling this concept of muscle atrophy? How many less subjective opportunities would arise? How would classic BoNT-A injections techniques would be impacted?	We are sure that understanding BoNT-A as a muscle atrophy tool for aesthetic purposes will bring us to new readings of previous articles and shed new light on future treatments.

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
