# Peer review of "High Precision Use of Botulinum Toxin Type A (BONT-A) in Aesthetics Based on Muscle Atrophy, Is Muscular Architecture Reprogramming a Possibility? A Systematic Review of Literature on Muscle Atrophy after BoNT-A Injections"

_toxins, 2022, doi:10.3390/toxins14020081_

Round 1

Reviewer 1 Report

The authors conducted a systematic review of literature on muscle atrophy after BoNT-A injections. The main conclusion was that seriated or single BoNT-A muscle injections can cause real muscle atrophy in a short or long-term basis in animals and humans. Muscular architecture reprogramming is a possible new approach in aesthetics. I have the following comments and suggestions.

  1. Figure 1: The word size in four circles should be the same. The word size in circle 1 is too small to be seen. In addition, the complete form of abbreviation of HC and LC should be showed in the legends.
  2. Table 5 and table 6 look like the same. Why repeated?
  3. Line 129: Why the maximum global score was set to 26? The maximum global score was 23 shown in the table 7.
  4. There were too many blank columns in table 10, table 11, table 13, and table 14. Please consider to re-arrange these tables to become more readers friendly. For example, I suggested table 15 as a 11x2, not 11x9.
  5. Conclusion: line 342: It was strange to see the final sentence “Depressor facial muscles……….. and facilitate a lift effect” because no significant findings in the result section or discussion section to support the statement.

Author Response

Dear colleague.

In attention to your questions, follow our answers.

Thank you very much for your considerations.

1. Figure 1: The word size in four circles should be the same. The word size in circle 1 is too small to be seen. In addition, the complete form of abbreviation of HC and LC should be showed in the legends

1 – Figure 1 has been modified

2. Table 5 and table 6 look like the same. Why repeated?

2 – Table 5 belongs to METHODS and shows the questions used for the quality analysis of articles. Table 6 belongs to RESULTS and the answers for the quality analysis are shown. They are different and complementary tables.

3.Line 129: Why the maximum global score was set to 26? The maximum global score was 23 shown in the table 7.

3 –The maximum score is 26, because there are 13 questions. Each question can receive a grade of zero, one or two. If the 13 questions receive the maximum grade (two), the total maximum grade would be 13x2=26. Table 7 does not correspond to your question. Check it again, please. The maximum score is explained on the bottom of table 5 (where you can see that there are 13 questions with a maximum possible grade of 2 (two) for each one. Legend of table 6 show the maximum possible score 26. To avoid misunderstandings, we will put the word POSSIBLE (Maximum POSSIBLE global score) on legend of table 6 (LINE 145).

4.There were too many blank columns in table 10, table 11, table 13, and table 14. Please consider to re-arrange these tables to become more readers friendly. For example, I suggested table 15 as a 11x2, not 11x9.

4 – Tables 10, 11, 13, 14 and 15 are very complex, but they were designed to show the results in a systematic and comparable way and can’t be altered. The problem is that the originals we’ve sent are to be used as LANDSCAPES, but we do not know why they are shown as  PORTRAITS. Is it possible to put them as LANDSCAPES?  Another problem is that the originals were tabled, centralized, and well disposed, but now they have a different format.

5. Conclusion: line 342: It was strange to see the final sentence “Depressor facial muscles……….. and facilitate a lift effect” because no significant findings in the result section or discussion section to support the statement.

5 – The idea of the conclusion is correct and is described and explained in DISCUSSION (line 301-310) and in table 4. We agree that the term (depressor muscle and elevator muscle were not associated with the explanation). To solve this problem, we put  the terms DEPRESSOR FACIAL MUSCLES and ELEVATOR FACIAL MUSCLES  in the discussion part (lines 308-310).

Reviewer 2 Report

I read the MS entitled " High precision use of botulinum toxin type A (BoNT-A) in aesthetics based on muscle atrophy. Is muscular architecture re-programming a possibility? A systematic review of literature on muscle atrophy after BoNT-A injections" submitted for publication to Toxin as a Systematic review article.

The authors criticize the current view that BonT-A injection serves for the treatment of wrinkles, and focus on short- and long-term atrophy of skeletal muscle after denervation and neuromuscular paralysis induced by BonT-A block of neuromuscular transmission. Through a systematic review of the work in the literature, they identified 30 papers published before 2020, in which the relationship between BonT-A injection and muscle atrophy in animals and humans was analyzed.

In these studies, after single or multiple injections of BonT-A, the appearance of muscle atrophy is highlighted, the role of which must be considered in the evaluation of the aesthetic treatment to be performed. The authors, therefore, suggest an approach to the aesthetic treatment of wrinkles based on reprogramming of muscle architecture through the targeted induction of moderate muscle atrophy in specific muscles, while others must be spared to continue to exercise their antigravity action.

The MS is clear and well written. Very useful are the diagrams and tables inserted in the text even if sometimes the latter is difficult to read. I suggest for instance trying to reduce the number of columns, transposing columns with rows, reduce the text in the cells (Table 15 could be improved by transposing columns with rows ? Table 13 is for me too long. Could you reduce the text in the cells ?). Text definition in tables is not high also in the supplementary file. Zooming in reveals that text is not well defined

In my opinion, some relevant works that have appeared in the literature are not cited (e.g. Eleopra R, et al. Clinical duration of action of different botulinum toxin types in humans. Toxicon. 2020) and perhaps a more careful evaluation should be carried out, before publication, given the systemic review nature of the MS.

I would then deepen the problem of denervation atrophy in humans (duration and extent), through a comparative evaluation in the literature, of the forms of denervation atrophy for causes other than botulinum toxin (spinal motor neuron degeneration, nerve root and nerve lesions; for instance papers regarding facial nerve denervation and recovery) with special attention to acute denervation.

In conclusion, I consider the MS publishable on Toxins after minor revisions.

Author Response

Dear colleague.

In attention to your questions, follow our answers.

Thank you very much for your considerations.

The authors, therefore, suggest an approach to the aesthetic treatment of wrinkles based on reprogramming of muscle architecture through the targeted induction of moderate muscle atrophy in specific muscles, while others must be spared to continue to exercise their antigravity action.

We do not suggest the treatment of wrinkles. We suggest the treatment of muscles based on the mechanism of action of botulinum toxin causing muscle atrophy. By doing that, we can have new perspectives of controlling the facial aging process targeting depressor facial muscles and preserving elevator facial muscles. Wrinkles improvements are not the objective of this new approach.

The quality of the texts of the tables and their presentation are jeopardized by the way they were modified by the editing process. Our originals are not PORTRAIT tables, but LANDSCAPES. Is it possible to maintain the originals? The tables were designed in a systematic and comparable way, that’s why they are too long sometimes and with blank spaces. The idea is to read them in LANDSCAPE and compare.

About the suggest paper: Eleopra R, Rinaldo S, Montecucco C, Rossetto O, Devigili G. Clinical duration of action of different botulinum toxin types in humans. Toxicon. 2020 May;179:84-91. doi: 10.1016/j.toxicon.2020.02.020. Epub 2020 Mar 14. PMID: 32184153.

We will consider using it in our next publications.

Round 2

Reviewer 1 Report

well revision